# Retrospective Study of the Upsurge of Enterovirus D68 Clade D1 among Adults (2014–2018)

**DOI:** 10.3390/v13081607

**Published:** 2021-08-13

**Authors:** Maxime Duval, Audrey Mirand, Olivier Lesens, Jacques-Olivier Bay, Denis Caillaud, Denis Gallot, Alexandre Lautrette, Sylvie Montcouquiol, Jeannot Schmidt, Carole Egron, Gwendoline Jugie, Maxime Bisseux, Christine Archimbaud, Céline Lambert, Cécile Henquell, Jean-Luc Bailly

**Affiliations:** 1Université Clermont Auvergne, LMGE CNRS 6023, UFR de Médecine et des Professions Paramédicales, 63001 Clermont-Ferrand, France; maxime.duval@uca.fr (M.D.); amirand@chu-clermontferrand.fr (A.M.); gwendoline.jugie@uca.fr (G.J.); mbisseux@chu-clermontferrand.fr (M.B.); carchimbaud@chu-clermontferrand.fr (C.A.); chenquell@chu-clermontferrand.fr (C.H.); 2CHU Clermont-Ferrand, Centre National de Référence Des Entérovirus et Parechovirus, Laboratoire de Virologie, 63003 Clermont-Ferrand, France; 3CHU Clermont-Ferrand, Service Des Maladies Infectieuses et Tropicales, 63003 Clermont-Ferrand, France; olesens@chu-clermontferrand.fr; 4CHU Clermont-Ferrand, Service de Thérapie Cellulaire et Hématologie Clinique, 63003 Clermont-Ferrand, France; jobay@chu-clermontferrand.fr; 5CHU Clermont-Ferrand, Service de Pneumologie, 63003 Clermont-Ferrand, France; dcaillaud@chu-clermontferrand.fr; 6CHU Clermont-Ferrand, Service de Gynécologie-Obstétrique, 63003 Clermont-Ferrand, France; dgallot@chu-clermontferrand.fr; 7Service de Réanimation, Centre Jean Perrin, 63003 Clermont-Ferrand, France; alautrette@chu-clermontferrand.fr; 8CHU Clermont-Ferrand, Centre de Référence et de Compétence Mucoviscidose, 63003 Clermont-Ferrand, France; smontcouquiol@chu-clermontferrand.fr; 9CHU Clermont-Ferrand, Service Des Urgences, 63003 Clermont-Ferrand, France; jschmidt@chu-clermontferrand.fr; 10CHU Clermont-Ferrand, Service de Pédiatrie Générale, 63003 Clermont-Ferrand, France; cegron@chu-clermontferrand.fr; 11CHU Clermont-Ferrand, Service Biométrie et Médico-Economie—Direction de la Recherche Clinique et Innovation, 63003 Clermont-Ferrand, France; clambert@chu-clermontferrand.fr

**Keywords:** enterovirus D68, respiratory conditions, adult patients, paediatric patients, next-generation sequencing, molecular epidemiology

## Abstract

Enterovirus D68 (EV-D68) has emerged as an agent of epidemic respiratory illness and acute flaccid myelitis in the paediatric population but data are lacking in adult patients. We performed a 4.5-year single-centre retrospective study of all patients who tested positive for EV-D68 and analysed full-length EV-D68 genomes of the predominant clades B3 and D1. Between 1 June 2014, and 31 December 2018, 73 of the 11,365 patients investigated for respiratory pathogens tested positive for EV-D68, of whom 20 (27%) were adults (median age 53.7 years [IQR 34.0–65.7]) and 53 (73%) were children (median age 1.9 years [IQR 0.2–4.0]). The proportion of adults increased from 12% in 2014 to 48% in 2018 (*p* = 0.01). All adults had an underlying comorbidity factor, including chronic lung disease in 12 (60%), diabetes mellitus in six (30%), and chronic heart disease in five (25%). Clade D1 infected a higher proportion of adults than clades B3 and B2 (*p* = 0.001). Clade D1 was more divergent than clade B3: 5 of 19 amino acid changes in the capsid proteins were located in putative antigenic sites. Adult patients with underlying conditions are more likely to present with severe complications associated with EV-D68, notably the emergent clade D1.

## 1. Introduction

Enterovirus D68 (EV-D68) is a re-emerging, pathogenic picornavirus that causes severe lower respiratory diseases in the paediatric population. It is associated with acute flaccid myelitis (AFM), a recently defined neurological illness [1]. The ability of EV-D68 to replicate within the upper and lower respiratory tracts is a key factor in the swift transmission of disease in the community, which results in large outbreaks [2,3,4]. EV-D68 illnesses have also been reported in healthy adults and those with underlying medical conditions, but the frequency of EV-D68 among different age groups has not been fully investigated. As for other enterovirus types, circulating strains of EV-D68 are genetically diverse and have been divided into four clades designated A–D [5,6]. The distribution of clades has been recorded over time and across countries but their frequency and the genetic and molecular variations among age groups remain elusive. To contribute to filling these knowledge gaps, we report the clinical characteristics of unselected patients with laboratory-confirmed EV-D68 respiratory disease. We also describe the changes that occurred at the genomes and proteins of EV-D68 clades reported in patient groups.

## 2. Materials and Methods

### 2.1. Patient Population, Study Design and Molecular Testing of Clinical Specimens

All patients with laboratory-confirmed EV-D68-associated disease and admitted to the University Hospital of Clermont-Ferrand from 1 June 2014 to 31 December 2018 were reviewed for the study. The respiratory specimens (nasopharyngeal swabs, nasopharyngeal aspirates, and broncho-alveolar lavages) were tested prospectively for enterovirus (EV) and rhinovirus (RV) with commercial RT-PCR assays, either RV/EV R-gene^®^ kit or FilmArray^®^ Respiratory Panel (bioMérieux, Marcy l’Etoile, France), as part of the routine screening for respiratory viral infections. When positive, the specimens were typed by sequencing the VP4/VP2 coding region or tested with an EV-D68 VP1 RT-PCR [7,8]. The complete or partial VP1 sequences of EV-D68-positive specimens were determined by in-house gene amplification and Sanger sequencing to assign a phylogenetic clade to the virus detected in each specimen. The VP1 sequences determined in this study were deposited under the accession numbers MT795858–MT795868. Data on patient characteristics, clinical manifestations, and co-morbidity factors were collected retrospectively frommedical charts, and they were recorded in an anonymized form for each patient and analysed. 

### 2.2. Construction of Metagenomic Libraries

Nucleic acids were extracted either from virus isolates at passage four (*n* = 8) or directly from clinical specimens (*n* = 27) with the NucliSENS^®^ easyMAG^®^ system (bioMérieux, Marcy l’Etoile, France). Virus isolation was performed in A549 cells (human Caucasian lung carcinoma) and the inoculated cultures were grown in a humidified 33 °C incubator with 5% CO_2_. Reverse transcription into DNA was performed with Superscript III^®^ (Invitrogen) and random or in-house primers for RNA extracted from virus isolates or clinical specimens, respectively. Two methods of amplification were used. The amplification of randomly synthetised cDNA was performed with overlapping in-house oligonucleotide primers and Phusion Flash^®^ master mix (ThermoFisher). PCR products were purified with NucleoSpin^®^ Gel and PCR Clean-up (Macherey-Nagel) before Sanger sequencing. The sequence of the amplicons obtained from virus isolates were determined with the Big Dye Terminator v.1.1 kit (Applied Biosystems) using the 3500 Dx Genetic Analyser^®^ system (Applied Biosystems). The amplification of specifically synthetised cDNA was performed with in-house primers (Appendix A) and the Platinum SuperFi enzyme (Invitrogen) to obtain 22/27 (2016, *n* = 4/4; 2018, *n* = 18/23) near full-length or whole genome amplicons. The amplicons were sequenced at the GENTYANE platform (INRAE, Clermont-Auvergne University, Clermont-Ferrand, France) by single molecule real-time sequencing with the Sequel I sequencer (Pacific Biosciences). The 3′ untranslated region was determined with an in-house amplification of the cDNA ends method. 

### 2.3. Sequence Datasets, Phylogenetic and in Silico Analyses

The 30 EV-D68 genomes determined in this study were grouped with those available in the GenBank database (as of 1 February 2020, *n* = 764 sequences). The open reading frame (ORF) sequences were used for the analyses (see below). A second dataset included 1433 whole VP1 gene sequences comprising 70 sequences from our patients, of which 32 were reported earlier [8,9,10]. Of the 38 VP1 sequences determined in this study, nine were obtained from the complete VP1 genes and 29 were derived from the whole genomes (Appendix A). The open reading frame (ORF) and VP1 nucleotide sequence datasets were subjected to phylogenetic analyses with the Nextstrain pipeline [11]. BEAST2 software was used for assessing divergence times of clades under explicit evolutionary models and computing the posterior probability distribution with a set of phylogenetic trees to assess node support [12]. The untranslated regions (UTRs) were analysed with datasets including 660 (5′UTR) and 628 (3′UTR) sequences. Phylogenetic analyses were performed with PhyML v3.0 using the maximum likelihood method and the Generalised Time-Reversible evolutionary model, and the statistical test aLRT SH-like [13]. The amino acid (ORF) and nucleotide (UTRs) sequences were compared to identify polymorphic and clade-specific positions within the B3 and D clades. The frequency of amino acid residues and nucleotides per site was determined using all EV-D68 clades and the relevant sites were plotted with Weblogo 3 [14]. Amino acid changes were mapped to 3D structures reported previously for the capsid proteins (Fermon strain, PDB code 4WM8) and the 3D polymerase (virus strain JPOC10-378, PDB code 5XE0) [15,16].

### 2.4. Statistical Analysis

Statistical analysis was performed with Stata software (v. 15; StataCorp, College Station, TX, USA). All tests were two-sided, with a Type I error set at 0.05. Categorical variables were expressed as frequencies and associated percentages, and the age as median and interquartile range. Comparisons between independent groups (children/adults) were made with the chi-square test or Fisher’s exact test for categorical variables and with the Mann–Whitney test for age.

## 3. Results

### 3.1. Clinical Characteristics of Patients

Routine diagnostic screening for RV/EV respiratory diseases was performed in 11365 respiratory samples from patients seen in the University Hospital of Clermont-Ferrand between 1 June 2014, and 31 December 2018. Of the 2101 respiratory specimens positive for RV/EV, 1836 (87%) were typed (Table 1). EV-D68 was detected in the specimens of 73 patients (median age 4 years [IQR 0.4–32.8]) of whom 61 (84%) were hospitalized. The 73 patients comprised 53 (73%) children (≤16 years; median age 19 years [IQR 0.2–4.0]; range 0.04–9.6) and 20 (27%) adults (>16 years; median age 53.7 years [IQR 34.0–65.7]; range 23.9–78.0). The number of laboratory-confirmed EV-D68 cases was stable in 2014, 2016, and 2018. In contrast, patient age varied with the proportion of adults increasing significantly from 12% (3/26) in 2014 to 22% (5/23) in 2016 and 48% (11/23) in 2018 (*p* = 0.01). The yearly proportions of respiratory specimens tested and subjected to enterovirus molecular typing in children and adults were stable in all three years.

Children were more likely to be hospitalised than adults (91% vs. 65%, *p* = 0.01; Table 2) and less likely to be admitted to an intensive care unit (ICU) (13% vs. 38%, *p* = 0.046). In our population, 71/73 (97%) patients had respiratory signs. Upper respiratory tract infections were the most frequent signs in both children (62%) and adults (45%). Twenty-seven (51%) children had clinical signs of asthma as against one (5%) adult. Chronic obstructive pulmonary disease (COPD) exacerbation was observed in five (25%) adults. Two clinical presentations were more frequent among adults with lung disease being reported in eight (40%) adults as against three (6%) children (*p* = 0.001) and neurological signs in seven (35%) adults compared to five (9%) children (*p* = 0.01). In adults, neurological signs were dominated by altered mental status (*n* = 5); sudden muscle weakness and cerebellitis were each observed in one adult. Of the 11 patients admitted to an ICU, nine (82%) had acute respiratory distress syndrome. There was one fatality reported in an infant presenting with congenital stridor and a co-infection with rhinovirus C.

67% (49/73) of patients had one or more comorbidity or risk factors for respiratory infections. In adults, the most frequent were chronic lung disease (12/20, 60%) and smoking (9/20, 45%). The other comorbidities in adults were diabetes mellitus (6/20, 30%) and chronic heart disease (5/20, 25%). Of the 13 adult patients admitted to hospital, 10 (77%) had chronic lung disease and four of them were admitted to an ICU. Five (25%) patients aged 33 to 70 years old required management in an ICU with a median (IQR) duration stay of 15 days (6–29). The four adult female patients who were pregnant did not require hospitalisation. No complication was reported at delivery and the new-borns were healthy. Asthma or wheezing or both were uncommon in adults (1/20, 5%) but were the most frequent underlying conditions in children (17/53, 32%; *p* = 0.02). 

Three children were co-infected with rhinovirus (*n* = 2) or coronavirus (*n* = 1). A microbiological screening including blood and expectoration cultures and urinary antigens (Legionella and Pneumococcus) was performed in 35 patients. Pneumococcus infection was detected in two hospitalised children and one hospitalised adult with COPD exacerbation, and colonisation with Pseudomonas and fungi was observed in two adult outpatients with cystic fibrosis.

### 3.2. Enterovirus Molecular Typing Data

Enterovirus molecular typing was performed prospectively by sequencing the VP4-VP2 genes in five (7%) specimens and the VP1 gene in 66 (90%) specimens (Table 2). The specimens of two other paediatric patients tested positive with an EV-D68-specific RT-PCR but the amplicons were not sequenced. The nucleotide sequences were used to assign a phylogenetic clade to the EV-D68 present in each specimen. In 2014, 18/26 (69%) patients were infected with clade B2, of whom all, but one, were children (Figure 1). The other eight patients were infected with clades D1 (*n* = 4; 2 adults), B1 (*n* = 3), and A (*n* = 1). Of the 23 patients recorded in 2016, 16 (70%) were infected with clade B3, of whom four (25%) were adults. One of the three non-clade B3 patients was an adult infected with clade D1; the other two were children infected with an EV-D68 whose clade was not determined. In 2018, B3 was the most frequent clade, affecting 13/23 (57%) patients, of whom five (39%) were adults. In the other 10 (43%) patients, of whom six (60%) were adults, EV-D68 was assigned to clade D1. Overall, clade D1 affected a higher proportion of adult patients (63%, *n* = 10/16) than clades B2 (6%, *n* = 1/18) and B3 (27%, *n* = 9/33) (*p* = 0.001). The proportion of patients who had an acute respiratory distress syndrome was higher in those infected with clade B2 (15/18, 83%) than those infected with clades B3 (10/33, 30%) or D1 (6/16, 38%) (*p* = 0.001). The five patients with COPD exacerbation had clade D1 infection (*p* < 0.001). Neurological signs were recorded in 12/73 (16%) patients, with a higher proportion in adults (*p* < 0.01). Altered mental status was reported in 5/20 (25%) adults: three were infected with clade B3, one with clade B2, and one with clade D1. Of the two adult patients infected with clade B3, one had signs of acute flaccid myelitis and the other of cerebellitis. Headaches were reported in two adults with clade D1 infection. Among the neurological signs reported in five children, irritability (*n* = 2) and hypotonia (*n* = 2) were associated with clades B3 and D1, respectively, and acute flaccid myelitis with clade B2 [10].

### 3.3. Analysis of Clade D1 Complete Genomes

The leftover respiratory specimens were available for 46/73 patients and were used to determine the complete EV-D68 genomes (Appendix A). The sequencing yield was better with the NGS approach directly from clinical specimens after amplification of the viral genome in one RT-PCR than with the Sanger method after cell culture (81% vs. 42%). The Ct values of the diagnostic RT-PCR available in 13 specimens that tested positive for EV-D68, ranged between 21.02 and 32.58. The mean sequencing coverage was 463 and 1767 full sequence reads, respectively, for the whole genome and near full-length amplicons. A phylogenetic tree was reconstructed for clade D (formerly designated A2) using eight genomes from this study and 28 genomes reported earlier in France and other countries (Figure 2A). The viral genomes in our patients were scattered within the D1 clade. Phylogenetic analysis of the larger VP1 sequence dataset confirmed this distribution pattern (Appendix AB). The close phylogenetic relationships of viruses in our and other French patients to those reported in neighbouring and distant countries were consistent with multiple introductions of EV-D68 during the same year and over years or with co-circulation of distinct lineages over time. The spread of the 2014 and 2018 EV-D68 lineages began shortly before our patients were infected. Sequence comparisons showed clade D-specific amino acid changes at 32 positions located throughout the open reading frame (Figure 2B). Six amino acid changes (including two insertions of two amino acid residues in the VP1 protein) were reported earlier [5,17]. 19/32 (59%) amino acid changes were detected in the capsid proteins, of which five were placed in putative immunoreactive epitopes, i.e., the VP1 BC and DE loops and the VP1 C-terminus [15,18]. In the D1 viruses detected in French patients, VP1 position 86 had different residues over time (an isoleucine in the 2014 lineage and a valine in the 2018 lineage). The sequential amino acid changes were confirmed by analysis of the VP1 dataset (Appendix AB). VP1 phylogeny showed two distinct clusters among the D1 sequences reported in Europe in 2018 both of which arose from amino acid changes at four positions, I1V, I5M, I86V, and I231V. The overall clade D1 differed from the other clades by 13 (42%) other amino acid changes, located in the non-structural proteins (Figure 2B). The 5′UTR of clade D1 differed by 27 nucleotide positions from that of the other clades of which six were in the secondary-structure domains II (*n* = 3), III, IV, and VI (Appendix A). Clade D1 was not different from the other clades in the 3′UTR (data not shown).

### 3.4. Analysis of Clade B3 Complete Genomes

The phylogeny of clade B3 assessed with complete genomes indicated that viral sequences reported among French patients in 2018 were distributed in two main clusters, a pattern that suggests virus co-circulation or independent virus introductions (Figure 3A). The overall B3 viruses had distinctive amino acid changes at 15 positions throughout the ORF (Figure 3B), four of which (VP4/18, VP2/151, VP3/241, and 3Cpro/55) were identified earlier [19]. The amino acid changes at six positions (VP2/74, VP3/163, VP1/206, 2C/102, 2C/277, and 3Cpro/50) arose in 2015–2016, and clearly differentiate clade B3 from all the other clades. The two 2018 phylogenetic clusters arose with amino acid changes at eight positions, D237E (VP3), S131N (VP1), T142S (2Apro), R264K, G273S, and A277T (2C ATPase), S59N (3A) and K166R (3Dpol). The first two changes (VP3/D237E and VP1/S131N) were also seen in clade D sequences. The second change was shared with the 2018 clade B3 viruses reported in the USA, which form a third cluster characterised by two additional amino acid changes in the VP1 BC loop (A83T and A86T). Clade B3 differed from the other clades by six nucleotide substitutions, one (U54C) in the 3′UTR and five in the 5′UTR (C107U, C188U (domain III), U654C, U678C, and A681G) (data not shown).

## 4. Discussion

This study shows that the prevalence of confirmed EV-D68 clade D1 was higher in adults than in children. Our study was made possible because since June 2014 all respiratory samples collected from patients of all ages admitted to or seen at the University Hospital of Clermont-Ferrand have been tested prospectively for enterovirus/rhinovirus, and all the positive cases have been tested for EV-D68. Despite a limited number of patients, this systematic evaluation enabled us to assess the frequency of EV-D68 in both children and adults, and together our data provide a consistent picture of virus transmission in both patient populations. The clinical presentations were dominated by respiratory symptoms, as previously reported [3,4,8]. EV-D68 was the most probable aetiological agent because viral and bacterial co-infections were infrequent in our patients. Upper respiratory tract infections and respiratory distress were reported in both paediatric and adult patients, but lung disease was more frequent in the adult group and only one case was associated with a co-infection with pneumococcus. Neurological signs were also more frequent in the adult patients and ranged from headache to signs of acute flaccid myelitis with muscle weakness of the lower limbs. In line with a previous observation [4], all adults had underlying conditions dominated by chronic respiratory diseases (60%), which were reported in 77% of the inpatient population. Respiratory comorbidities are recognized risk factors for severe infections with the respiratory syncytial virus, which frequently causes lung disease in adult patients [20]. Respiratory viruses such as rhinoviruses, respiratory syncytial virus, and influenza, are well-known exacerbation factors of COPD [21]. Like rhinoviruses, with which it shares common biological features [22], EV-D68 can act as a trigger for disease exacerbation in adults. 

In our study, 12 adults (60%) with an EV-D68 infection were females of whom four were seen as outpatients during pregnancy. This figure was not significant in our patient population, but sex disparities have been reported in respiratory diseases caused by other RNA viruses (influenza A virus, respiratory syncytial virus, and coronavirus) and been attributed in part to an effect of sex hormones on innate immune responses [23]. Hormone-triggered modulation of the expression of cellular receptors was reported to promote sex disparities in SARS coronavirus-2 infections [24]. Pregnancy is known as a risk factor for severe influenza A infections, possibly owing to immune suppression by elevated progesterone and oestrogen levels [23]. In our study, none of the four pregnant women had a severe respiratory infection. 

Our phylogenetic analyses showed that clades B3 and D1 arose globally at the same time (between 2009 and 2010) but the upsurge of adult cases in our population was recorded in 2018, a year during which clade D1 re-emerged. Our data are consistent with those collected by the French National Reference Centre, which indicated a large increase in the proportion of adults infected with EV-D68, from 12% in 2014 to 9% in 2016 and 45% in 2018. In 2018, 79% (*n* = 30/38) of adult patients who tested positive for EV-D68 were infected with clade D1 (https://cnr.chu-clermontferrand.fr/CNR (accessed on 1 February 2020)). Previous studies suggested a predominance of the EV-D68 clade D1 in adults [17,25,26,27,28]. Analysis of our global dataset, which included VP1 sequences of worldwide origin and information on age, showed that 68% (*n* = 67/99) of clade D1 sequences were seen in respiratory samples of adults, which is consistent with a possible link between patient age and virus clade. These findings observed across countries suggest a high frequency of clade D1 among adults with an EV-D68 infection. However, our data show a twofold increase in the proportion of adults infected with EV-D68 clade B3 between 2016 (20%) and 2018 (38%), which indicates that the emergence of clade D1 is not the only factor associated with the upsurge of disease cases in adults. Overall, comorbidities or risk factors could account for the spread of EV-D68 within the adult population. 

To examine the potential role of virus-associated factors in the emergence of EV-D68 in the adult patients, we expanded the number of full-length EV-D68 genomes from France and investigated the variations in viral proteins between clades B3 and D1. 59% of the 32 mutations found in clade D1 were in the capsid proteins but none occurred among the sites reported to bind sialic acid, the cell receptor reported for EV-D68 [15]. Five mutations arose in the epitopes displayed on the VP1 protein, which include the polymorphic position 86. In this protein, two additional variations could have an impact on the capsid surface, the deletion of an amino acid residue within the DE loop (position 128) and the insertion of two residues in the C-terminus (positions 295–296). The DE and BC loops are possible epitopes of EV-D68, and the C-terminus was recently described as an immunoreactive structure [15,18]. The amino acid changes found in clade D1 could lower cross-protection in adults by decreasing binding of antibodies generated in response to an earlier infection. Using sera collected in 2006 and 2016, two European studies showed that the neutralising antibody seroprevalence and geometric mean titres (GMT) to EV-D68 clade B3 increase with age [29,30]. Above the age of 20 years, which fits the characteristics of the adult population in our study, the seropositivity rate approaches 100% and is associated with high GMTs. Accordingly, it seems plausible that the adult patients of our study have been infected with an EV-D68 clade B3. Although evidence of cross-reactive immunity is not firmly established between EV-D68 clades, Karelehto and colleagues [29] showed that sera collected in 2006 and 2007 had high neutralising titres to the clade B3 CF183054 virus strain (isolated in the present study; Appendix A), while in this study, we dated the emergence of this clade to 2009. In another study [31], adult sera collected in 2012 and 2013 (Kansas City population) neutralised the clade D1 virus KM851231-USA-2014 (see Figure 2A) less efficiently than clades B1 and B2 viruses, and the Fermon reference strain. The amino acid changes in the surface of clade D1 viruses could reduce cross-clade immunity and foster re-infection of adults, thereby explaining the increase in the adult case numbers in 2018 while D1 and B3 were in co-circulation. Gilrane and colleagues [2] recently reported clade D1 emergence during the 2018 outbreak in USA. These viruses are very close relatives of clade D1 viruses reported in our study and could have a similar epidemiological impact. 

Selection bias in patient population can be excluded from our study, whose main limitations arise from its retrospective design. First, patient follow-up was not recorded. Acute exacerbation COPD caused by rhinovirus infection was associated with lung function decline and disease progression [21]. Accordingly, re-assessment of patients with underlying conditions would provide important clinical data on potential long-term respiratory sequelae after EV-D68 disease. Second, the retrospective collection of medical charts could have provided incomplete clinical data in some patients, and third, the origin of EV-D68 infections in the adult patients could not be investigated. Thus, an epidemiological survey was not possible to assess how the adults acquired their infection and whether they were in contact with sick children within familial or professional settings. 

## 5. Conclusions

Widespread surveillance and routine screening of children and adults with comorbidities are essential to fill the gaps in knowledge about the dynamics of EV-D68 transmission between the different age groups. Health authorities and clinicians should be aware of the potentially severe complications in adults with underlying medical conditions arising from EV-D68 infection during periods of epidemic transmission. Systematic analysis of viral whole-genome data is important to trace the origin of EV-D68 outbreaks, investigate molecular drivers of virus transmission, and support evidence-informed public health decision.

## Figures and Tables

**Figure 1 viruses-13-01607-f001:**
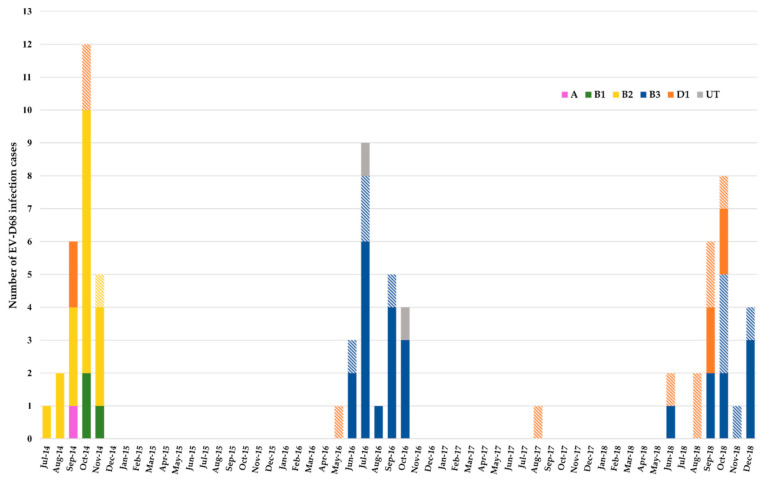
Monthly distribution of EV-D68 infections, Clermont-Ferrand, 2014–2018. Histogram colours match the EV-D68 clade related to the infection case as indicated on the graph. Hatched colours match EV-D68 in adult patients. Solid colours match EV-D68 in paediatric patients. UT: untyped.

**Figure 2 viruses-13-01607-f002:**
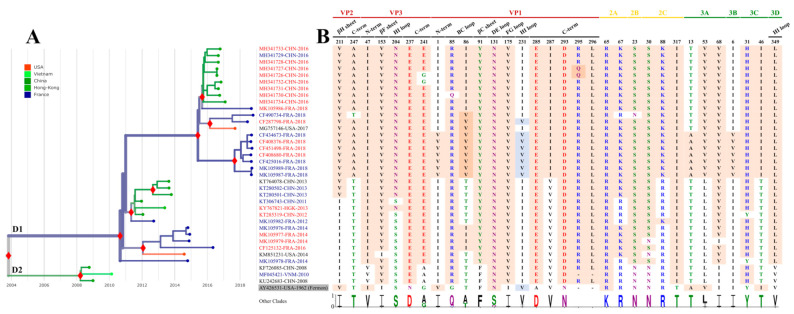
Phylogeny of EV-D68 clade D and molecular variation of viral proteins. (**A**) The phylogenetic tree was obtained with the Nextstrain pipeline using a global dataset of 734 EV-D68 whole genomes available in Genbank (as of 1 February 2020) and 30 new sequences obtained in this study. The solid red diamonds indicate statistical support of main nodes (posterior probability ≥0.98). The colours of sequence names match with age groups of related patients: red and blue colours match with adults (>16 years) and children (≤16 years), respectively. Sequence names in black had no corresponding information on related patient age. The branches are coloured according to the geographic origin. (**B**) Identification of protein positions having amino acid residues specific to clade D. The amino acid positions from other EV-D68 clades are expressed in percentages on the graph with Weblogo 3 [14]. Amino acids with pink background indicate specific positions. Amino acids with blue background match unspecific positions. Amino acids are coloured according to their chemistry.

**Figure 3 viruses-13-01607-f003:**
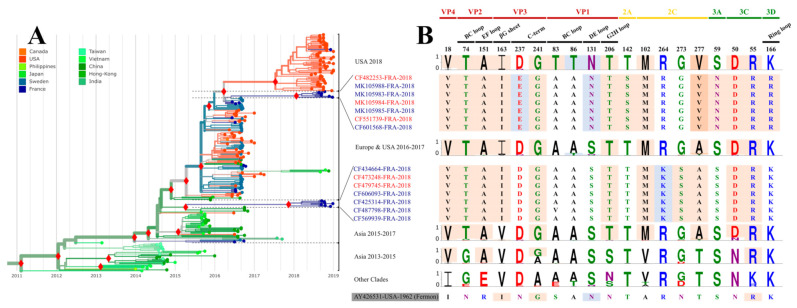
Phylogeny of EV-D68 clade B3 and molecular variation of viral proteins. (**A**) The phylogenetic tree was obtained with the Nextstrain pipeline using a global dataset of 734 EV-D68 whole genomes available in Genbank (as of 1 February 2020) and 30 new sequences obtained in this study. The solid red diamonds indicate statistical support of main nodes (posterior probability ≥0.98). The colours of sequence names match with age groups of related patients: red and blue colours match with adults (>16 years) and children (≤16 years), respectively. Sequence names in black had no corresponding information on related patient age. The branches are coloured according to the geographic origin. (**B**) Identification of protein positions having amino acid residues specific to clade B3. The amino acid positions from other EV-D68 clades are expressed in percentages on the graph with Weblogo 3 [14]. Amino acids with pink background indicate specific positions. Amino acids with blue background match unspecific positions. Amino acids are coloured according to their chemistry.

**Table 1 viruses-13-01607-t001:** Yearly screening for rhinovirus/enterovirus (RV/EV) in respiratory samples from patients presenting to the University Hospital, Clermont-Ferrand, France.

Data ^1^	Age Group ^2^	2014 ^3^	2015	2016	2017	2018	Total
Total samples tested	Adults	397 (55%)	1043 (58%)	2201 (70%)	1678 (59%)	1809 (63%)	7128 (63%)
	Children	324 (45%)	752 (42%)	948 (30%)	1153 (41%)	1060 (27%)	4237 (27%)
Positive detection of RV/EV	Adults	55 (14%)	103 (10%)	208 (9%)	134 (8%)	209 (12%)	709 (10%)
	Children	138 (43%)	225 (30%)	332 (35%)	335 (29%)	362 (34%)	1392 (33%)
Enterovirus typing	Adults	46 (84%)	91 (88%)	188 (90%)	121 (90%)	191 (91%)	637 (90%)
	Children	127 (92%)	183 (81%)	289 (87%)	276 (82%)	324 (90%)	1199 (86%)
EV-D68 typing	Adults	3 (12%)	0	5 (22%)	1 (100%)	11 (48%)	20 (27%)
	Children	23 (88%)	0	18 (78%)	0	12 (52%)	53 (73%)
	Median age	4 (0.7–6.9)		3.1 (0.2–7.6)		7.8 (0.5–54)	4 (0.4–32.8)
	Total	26	0	23	1	23	73

^1^ Data are *n* (%) or median (IQR). ^2^ Adults, patients aged >16 years; children, patients aged 0–16 years. ^3^ In 2014, the data were recorded from June 1st.

**Table 2 viruses-13-01607-t002:** Demographic and clinical characteristics of patients presenting to the University Hospital, Clermont-Ferrand with enterovirus D68-associated diseases between 1 June 2014, and 31 December 2018.

Characteristics ^1^	All Patients	Children	Adults	*p* Value
Total number of patients	73	53 (73%)	20 (27%)	
Age (years)	4.0 [0.4–32.8]	1.9 [0.2–4.0]	53.7 [34.0–65.7]	<0.001
Sex (male)	41 (56%)	33 (62%)	8 (40%)	0.09
Hospital admission	61 (84%)	48 (91%)	13 (65%)	0.01
ICU admission	11/61 (18%)	6/48 (13%)	5/13 (38%)	0.046
Fever	36 (49%)	25 (47%)	11 (55%)	0.55
Respiratory signs	71 (97%)	51 (96%)	20 (100%)	1.00
Upper respiratory tract infection	42 (58%)	33 (62%)	9 (45%)	0.18
Asthma	28 (38%)	27 (51%)	1 (5%)	<0.001
Bronchitis/Bronchiolitis	10 (14%)	10 (19%)	0 (0%)	0.053
Lung disease	11 (15%)	3 (6%)	8 (40%)	0.001
COPD exacerbation	5 (7%)	0 (0%)	5 (25%)	0.001
Acute respiratory distress	34 (47%)	28 (53%)	6 (30%)	0.08
Ear-Nose-Throat signs	2 (3%)	2 (4%)	0 (0%)	1.00
Neurological signs	12 (16%)	5 (9%)	7 (35%)	0.01
Digestive signs	9 (12%)	5 (9%)	4 (20%)	0.25
Cardiac signs	3 (4%)	2 (4%)	1 (5%)	1.00
Multiorgan failure	1 (1%)	0 (0%)	1 (5%)	0.27
All underlying conditions	49 (67%)	29 (55%)	20 (100%)	<0.001
History of asthma/wheezing	18 (25%)	17 (32%)	1 (5%)	0.02
Chronic lung disease	15 (21%)	3 (6%)	12 (60%)	<0.001
Chronic heart disease	7 (10%)	2 (4%)	5 (25%)	0.01
Chronic kidney disease	1 (1%)	0 (0%)	1 (5%)	0.27
Chronic liver disease	1 (1%)	0 (0%)	1 (5%)	0.27
Diabetes mellitus	7 (10%)	1 (2%)	6 (30%)	0.001
Immunocompromising conditions	4 (5%)	2 (4%)	2 (10%)	0.30
Neurological disorders	1 (1%)	0 (0%)	1 (5%)	0.27
Current tobacco use	9 (12%)	0 (0%)	9 (45%)	-
Current alcohol overuse	4 (5%)	0 (0%)	4 (20%)	-
Obesity	3 (4%)	0 (0%)	3 (15%)	0.02
Pregnancy	4/32 (12%)	0/20 (0%)	4/12 (33%)	-
Other underlying conditions ^2^	6 (8%)	6 (11%)	0 (0%)	0.18
EV-D68 clade A	1 (1%)	1 (2%)	0 (0%)	-
EV-D68 clade B1	3 (4%)	3 (6%)	0 (0%)	-
EV-D68 clade B2	18 (25%)	17 (32%)	1 (5%)	-
EV-D68 clade B3	33 (45%)	24 (45%)	9 (45%)	-
EV-D68 clade D1	16 (22%)	6 (11%)	10 (50%)	0.001 ^4^
Clade not determined ^3^	2 (3%)	2 (4%)	0 (0%)	-

^1^ Data are *n* (%) or median [IQR]. Data collected retrospectively from medical charts of patients. ^2^ Congenital stridor (*n* = 2), premature (*n* = 2) one of which was associated with bronchopulmonary dysplasia, Gordon syndrome (*n* = 1), and stenosis of two pulmonary veins (*n* = 1). ^3^ Enterovirus D68 was genotyped with an enterovirus D68-specific RT-PCR, but the amplicons were not sequenced. ^4^ The statistical analysis compared the proportion of adults infected by the clades B2, B3, and D1; the data for the other two clades were not included. Abbreviations: ICU = intensive care unit. COPD = chronic obstructive pulmonary disease.

## Data Availability

The genome sequences have been deposited to GenBank under the accession numbers MT789734–MT789755 and MT791927–MT791934.

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
