# Peer review of "Retrospective Study of the Upsurge of Enterovirus D68 Clade D1 among Adults (2014–2018)"

_viruses, 2021, doi:10.3390/v13081607_

Round 1

Reviewer 1 Report

The manuscript “Retrospective study of the upsurge of enterovirus D68 clade D1 2 among adults (2014–2018)” by Duval et al presents the retrospective molecular epidemiology of the enterovirus D68 (EV-D68) circulating in Adults for a period of 4.5 years and identified in the hospital of Clermont-Ferrand (France). Authors have first described the studied adult population by contrasting it with children. Then they propose statistically significant clinical risk factors associated with disease severity. In a second time, the phylogenetic analyses explore the clades associated to the circulating EV-D68, as well as their evolution over time and the amino acid changes. This manuscript is a nicely written, easy to understand. Therefore, this manuscript needs some revisions.

Major comments

Phylogenetic trees (figures 2, 3, S1, and S2), their legends, as well as the material and method do not indicate how (or if) the node consistencies have been determined, and if the nodes are consistent. Which, thus, doesn’t allow to appreciate the significance of the phylogenetic trees.

In the material and method, it is mention that 38 VP1 sequences have been realized in this study (Line 101-102), however the deposited VP1 sequence accession number MT795858-MT795868 (line 74) would indicate only 10 sequences submitted on databases. Authors should clarify that point in the main text, as all sequence accession numbers are summarized in Table S2.

In figure 1, the logic of the colors should be simplified. Indeed, it is a bit confusing between B2 subclade in yellow with D1 subclade in orange, or else B3 in dark blue with A in purple, and with Adults in light colors versus children probably in dark colors.

Minor comments.

Table 1: the percentage of total samples tested for children (despite it is obvious) is missing.

Table S1: the addition of the genomic coordinates for the primers would be helpful.

Line 55: the brackets of the references 5 and 6 are missing in the text

Author Response

Replies to the Reviewers’ comments – Reviewer 1

1) Phylogenetic trees (figures 2, 3, S1, and S2), their legends, as well as the material and method do not indicate how (or if) the node consistencies have been determined, and if the nodes are consistent. Which, thus, doesn’t allow to appreciate the significance of the phylogenetic trees.

Reply. We agree with the reviewer and we have included the requested data in the section materials and methods (on page 3, lines 108 – 109, and line 113) and the legends to the figures. We also have indicated node support on the figures.

2) In the material and method, it is mention that 38 VP1 sequences have been realized in this study (Line 101-102), however the deposited VP1 sequence accession number MT795858-MT795868 (line 74) would indicate only 10 sequences submitted on databases. Authors should clarify that point in the main text, as all sequence accession numbers are summarized in Table S2.

Reply. We agree with reviewer’s comment and we have clarified the distribution of the 38 VP1 sequences in the main text (page 3, lines 103 – 105) and the note to the supplementary Table 2.  

3) In figure 1, the logic of the colors should be simplified. Indeed, it is a bit confusing between B2 subclade in yellow with D1 subclade in orange, or else B3 in dark blue with A in purple, and with Adults in light colors versus children probably in dark colors.

Reply. We have changed the colours in the figure. The data regarding adults were indicated with hatched colours.

Minor comments.

Table 1: the percentage of total samples tested for children (despite it is obvious) is missing.

Reply. Done.

Table S1: the addition of the genomic coordinates for the primers would be helpful.

Reply. Done.

Line 55: the brackets of the references 5 and 6 are missing in the text

Reply. Done.

Reviewer 2 Report

In this paper, Duval and colleagues conducted a retrospective study of all EV-D68-positive patients including adults in 2014-2018 and analyzed full-length EV-D68 genomes for the predominant clades B3 and D1. Although the number of EV-D68-positive cases was limited (and stable) in 2014-2018, the proportion of adults was increased from 12% in 2014 to 22% in 2016 and 48% in 2018. The prevalence of new EV-D68 clade D1 was higher in adults than in children, particularly in 2018. This study highlights critical concern on the potential risk of EV-D68 infection, including severe respiratory infections and neurological complications even in elder age groups. As the authors mentioned, epidemiological and genetic analysis of EV-D68 presented in this paper would be worthwhile to monitor the epidemiological and clinical settings of EV-D68 clade D1 infections in Europe, and the methodological approaches applied in this study are convincing.

Therefore, I just mention several concerns to be addressed or clarified as bellow.

Specific comments

  1. Page 2; More details on virus isolation (cell lines, culture and temp conditions, etc.) should be described. Are there any differences in the dominant EV-D68 genetic clade between virus isolates and direct genome amplification?
  2. The authors should review and discuss about the available seroprevalence reports among different age groups in France or Europe. Are the EV-D68-positve adult patients in this study expected to be seronegative individuals or second or more EV-D68 infections?
  3. Page 10, line 362; vulnerable adults, immunologically naïve or persons with underlying disease?
  4. Any evidence of genetic recombination between different genetic clades?

Minor points

 line 55; 5,6 to be [5,6]

Table layout should be arranged more appropriately.

Author Response

Replies to the Reviewers’ comments – Reviewer 2

1) Page 2; More details on virus isolation (cell lines, culture and temp conditions, etc.) should be described. Are there any differences in the dominant EV-D68 genetic clade between virus isolates and direct genome amplification?

Reply. We have included details on virus isolation (on page 2, lines 81 – 83). We have had more difficulties to isolate virus strains assigned to enterovirus D68 clade D (year 2018) than clade B.

2) The authors should review and discuss about the available seroprevalence reports among different age groups in France or Europe. Are the EV-D68-positve adult patients in this study expected to be seronegative individuals or second or more EV-D68 infections?

Reply. Others or we did not performed serological study in France. We have reviewed and discussed the serological data in 2 European studies (pages 9 and 10, lines 354 – 363). We also have included an additional reference (no. 30). From the available data, we expect that in our study, the adult patients tested positive for EV-D68 were seropositive individuals.

3) Page 10, line 362; vulnerable adults, immunologically naïve or persons with underlying disease?

Reply. We have changed “vulnerable adults” to “adults with underlying medical conditions”.

4) Any evidence of genetic recombination between different genetic clades?

Reply. We have found no evidence of recombination in our sequence dataset.

5) line 55; 5,6 to be [5,6]

Reply. Done.

6) Table layout should be arranged more appropriately.

Reply. We will do the requested changes during edition processing.

Round 2

Reviewer 1 Report

/